🔓 | **Open Peer Review** | Virology | Research Article

# Diagnostic performance of upper airway sampling sites for SARS-CoV-2 and influenza testing

Mary Lopez-Perez,[1] Thomas Benfield,[2,3] Kathrine K. Jakobsen,[4] Mette Hyldig Dal,[4] Sabrina Dandanell Stange,[4] Annette Kjær Ersbøll,[5,6] Helene Larsen,[7] Sanne Schou Berger,[7] Tobias Gredal,[5] Christian von Buchwald,[2,4] Nikolai Kirkby,[8] Tobias Todsen[2,4,9]

**ABSTRACT** Early diagnosis of upper respiratory infections is essential to control infectious disease transmission within the community and to initiate relevant antiviral treatments. Nonetheless, variable sensitivities of sampling sites are often overlooked. We conducted a clinical trial at a COVID-19 outpatient test center, where healthcare workers collected nasopharyngeal, throat, nasal swabs, and saliva specimens. Each specimen was tested by RT-PCR for severe acute respiratory syndrome coronavirus 2 (SARS-CoV-2), influenza virus A and B, and respiratory syncytial virus (RSV). Clinical information was collected at enrollment and again 13 monthslater using an online questionnaire. From 4 March to 31 March 2023, 253 individuals were enrolled. Data from 250 participants were included in the analysis. SARS-CoV-2 was the most frequent viral infection (48%), followed by influenza A or B (5%), and a combination of SARS-CoV-2 and influenza (9%). RSV was not detected in any specimen. Several participants carried two or three viruses simultaneously. Throat swabs were significantly more sensitive for detecting SARS-CoV-2 (79%) and influenza (64%) than samples from other sites. In contrast, saliva had the lowest sensitivity for SARS-CoV-2 (43%) and was unsuitable for detecting influenza. The sensitivity for SARS-CoV-2 (88%) and influenza (100%) improved when results from throat and nasal swabs were combined. Throat swabs were more sensitive than nasopharyngeal swabs and saliva for molecular detection of SARS-CoV-2 and influenza. A combined throat and nasal swab is highly recommended to increase test sensitivity in patients presenting with upper respiratory infection symptoms.

**IMPORTANCE** Upper respiratory infections are the most common condition in primary care. Therefore, their early diagnosis is essential to control infectious disease transmission within the community. Here, we show that throat swabs were more sensitive than nasopharyngeal swabs and saliva for detecting severe acute respiratory syndrome coronavirus 2. Furthermore, throat and nasal swabs were more effective in detecting influenza compared to nasopharyngeal swabs.

**CLINICAL TRIALS** Registered at ClinicalTrials.gov NCT05765838.

**KEYWORDS** diagnosis, influenza, SARS-CoV-2, upper respiratory tract

Upper respiratory infections (URIs) are the most common presenting condition in primary care globally (1). Although most URIs are caused by viruses that typically result in mild and self-limiting symptoms, pathogens like severe acute respiratory syndrome coronavirus 2 (SARS-CoV-2), influenza viruses, and respiratory syncytial virus (RSV) may progress to more severe acute respiratory infections. Seasonal influenza accounts for around 1 billion cases annually, including 35 million casesof severe illness, hospitalizations, and deaths (2). On the other hand, SARS-CoV-2 has caused more than 8 million reported deaths worldwide since late 2019 (3).

**Peer Reviewer** Matthew Martin Hernandez, Icahn School of Medicine at Mount Sinai, New York, New York, USA

Address correspondence to Tobias Todsen, Tobias.Todsen@regionh.dk.

T.B. reports grants from Novo Nordisk Foundation, Lundbeck Foundation, Simonsen Foundation, GSK, and Pfizer; personal fees from GSK, Pfizer, Bavarian Nordic, Gilead, MSD, Janssen, Moderna, Shionogi, and Astra Zeneca; outside the submitted work. T.T. reports grants from Novo Nordisk Foundation and personal fees from Johnson and Johnson. The other authors declare that they have no conflicts of interest.

See the funding table on p. 9.

Early diagnosis is essential for controlling infectious transmission within the community and starting antiviral treatment for patients at risk of progressing to severe acute respiratory infection. High-quality samples are essential, but the choice of anatomical sites for sampling usually depends on local preferences (4, 5) and less on sampling site performance and cost (4–6). Nasopharyngeal swabs (NPS) are considered the gold standard for URI testing (4, 7, 8). However, studies during the COVID-19 pandemic have shown that SARS-CoV-2 may be detected in the oropharynx before the nasal cavity during the early stages of infection (6, 9–11). We therefore conducted a clinical trial to compare head-to-head molecular sensitivity for SARS-CoV-2, influenza virus A and B, and respiratory syncytial virus in nasopharyngeal, throat, and nasal swabs, as well as saliva.

## MATERIALS AND METHODS

### Study design and participants

We conducted a randomized clinical trial at a public COVID-19 test center in Valby, Copenhagen, Denmark, from 4 March to 31 March 2023 (Fig. 1). All adults requesting a free-of-charge SARS-CoV-2 RT-PCR test for diagnosis and screening purposes during this period were invited to participate in the study. Sample size was not determined. To ensure we also had data on potentially non-infected participants, we invited healthy healthcare professionals to participate in the study. Exclusion criteria included medical conditions preventing regular swab sampling (e.g., tracheostomy, laryngectomy, or

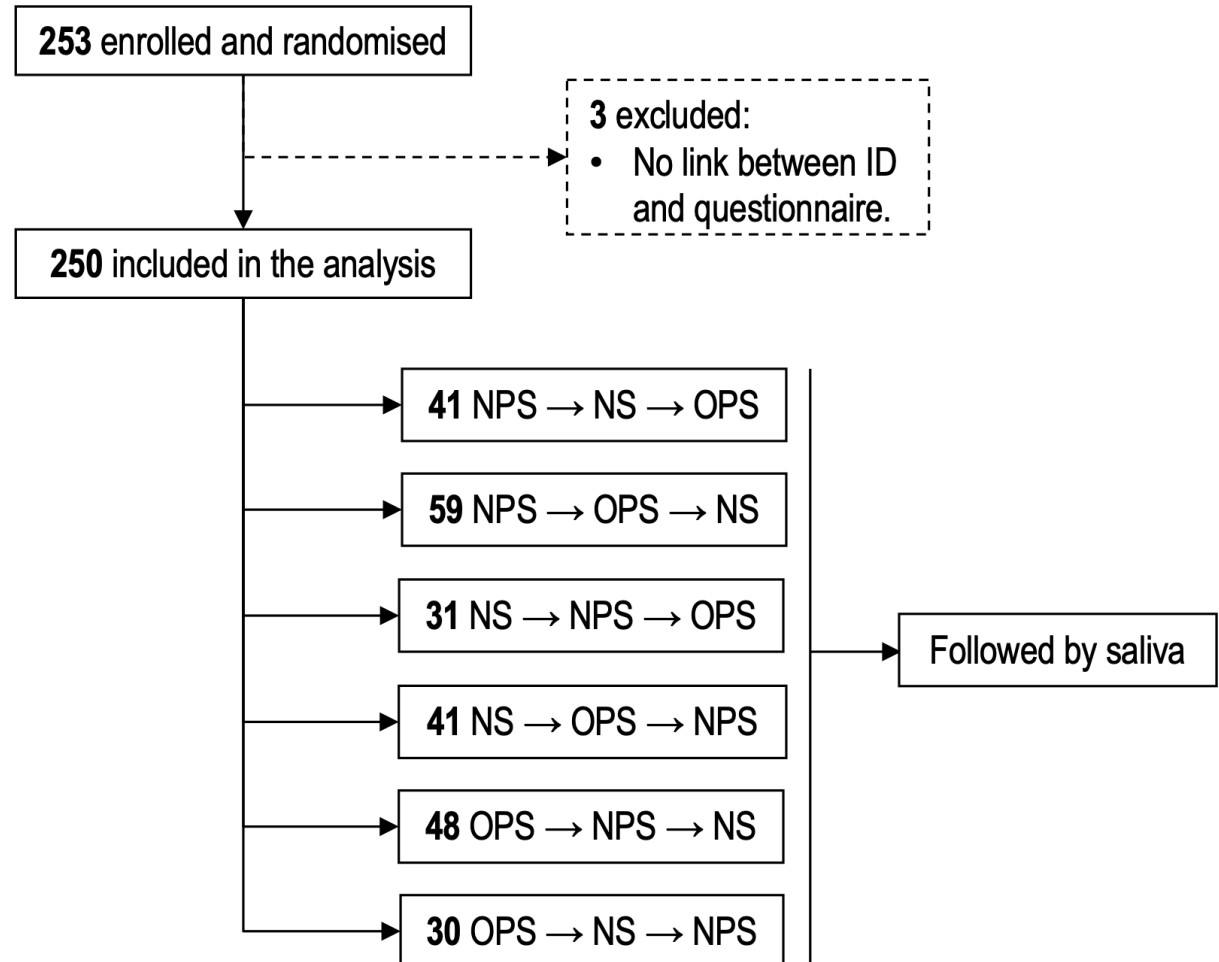

**FIG 1** Study flow chart.

oropharyngeal cancer surgery) and participants who did not understand written and spoken Danish. Enrolled participants had nasopharyngeal, throat, and nasal swabs, as well as saliva specimens collected according to their randomization to ensure equal comparison of specimen types. The specimens were collected by healthcare workers who received competence-based training in the different swabbing techniques (12). After enrollment, and again 13 monthslater, participants received a link to an online questionnaire to collect information on the type and duration of symptoms, vaccination status, comorbidities, and disease development.

Nasopharyngeal, throat, and nasal swabs were collected randomly, followed by saliva collection from all participants (Fig. 1). The randomization list was incorporated into the REDCap Randomization Module, and participants were assigned to each group during enrollment and registration at the test center.

## Sample collection

Upper respiratory specimens were collected by trained healthcare workers using nylon-flocked swabs (Meditec A/S, Helsingør, Denmark) as previously described (6, 13, 14). Briefly, a throat swab (TS) was collected from the posterior oropharyngeal wall and palatine tonsils, avoiding the tongue and cheeks. The nasal swab (NS) was inserted approximately 2–4 cm into one nostril and rotated thrice before withdrawal. The procedure was then repeated in the other nostril. For the nasopharyngeal specimen (NPS), the swab was inserted into the nostril until resistance was felt at the posterior pharynx, rotated three times, and withdrawn slowly with a rotating motion (15). Swabs were placed into separate sterile tubes containing 2 mL of viral transport media (Wuxi NEST Biotechnology). Participants were also instructed by the healthcare workers to collect at least 2 mL of saliva in a 50 mL collection tube by using the drooling technique (13). Then, 200 µL of saliva was mixed with 80 µL lysis buffer and incubated for 10 min at room temperature. All specimens were stored at −70°C until further analysis at the Technical University of Denmark (DTU), Lyngby, Denmark.

## RT-PCR analysis

RT-PCR analysis of samples was performed at DTU, testing each specimen for SARS-CoV-2, influenza A, influenza B, and RSV. Nucleic acid was extracted using an RNAdvance viral isolation kit, based on SPRI paramagnetic bead-based technology, on a Biomek i7 liquid handler (Beckman Coulter), and 5 µL of extracted RNA was used in the PCR reactions. SARS-CoV-2 RNA was detected using a multiplexed version of the CDC N-gene one-step RT-PCR, targeting two N-gene segments and the human RNase P ribozyme gene (RNase P) to assess the presence of human genetic material and verify the quality of the sampling procedure (CoviDetect COVID-19 multiplex RT-qPCR assay, Pentabase A/S, Denmark). RT-PCR tests were considered SARS-CoV-2 positive if the cycle threshold (Ct) was 35 or lower for one or two target gene segments and below 28 for the RP gene. This cutoff was determined using serial dilutions of known quantities of viral RNA in water. Influenza A, influenza B, and RSV were detected using the dispense-ready RespiDetect Respiratory Panel 1 RT-qPCR assay (Pentabase A/S, Denmark), analyzed on the BaseTyper 48.4 (Pentabase A/S, Denmark). Positive samples in Cy5 (Ct < 33) at both 56°C and 72°C were considered positive for influenza B, positive samples in FAM (Ct < 34) were considered positive for influenza A, and samples positive in Texas Red (Ct < 34) were considered positive for RSV. It is possible for a sample to test positive for more than one virus. The RNase P (HEX) was used to assess if the sampling was conducted correctly (Ct < 34).

## Statistical analysis

Study data were collected and managed using REDCap (Nashville, TN, USA). GraphPad Prism version 10.2.2 (GraphPad Software, San Diego, CA, USA) and Statistical Analysis Software version 9.4 (SAS Institute, North Carolina, USA) were used for the statistical

analyses. A specimen was classified as positive based on the Ct values reported above. For participant classification, molecular test results for each specimen were added, meaning that a participant was considered positive for a specific virus if any of their specimens tested positive. A participant was classified as negative if all tests were negative. Influenza infection was defined as having influenza virus A, B, or both in the same or different specimens. Co-infection with SARS-CoV-2 and influenza was defined as participants testing positive for both in the same or different specimens. The combined results of nasopharyngeal, throat, nasal swabs, and saliva tests were used as the diagnostic reference to calculate sensitivity, positive predictive value, and negative predictive value (NPV) for diagnosis in each specimen. A logistic regression analysis using generalized estimating equations was used to compare the sensitivity of SARS-CoV-2 and influenza among specimens. The 95% confidence intervals were calculated with adjustments to account for the order in which specimens were collected (randomization). The agreement of the results between the paired nasopharyngeal and throat, nasal swabs, or saliva specimens was assessed separately for each respiratory virus using the kappa coefficient. The number of samples is reported in each figure. A P value <0.05was considered statistically significant.

## RESULTS

From 4 March to 31 March 2023, 253 individuals were enrolled in the study (Fig. 1), but three participants were excluded because their results could not be linked to their corresponding identification number and questionnaire. The final analysis involved 250 individuals (129 [52%] women; median age, 45 years [IQR: 33–57 years]), including 35 healthy individuals who were invited to participate in the study. Characteristics of participants are presented in Table 1. Among all participants, 201 (80%) presented with at least one symptom, and in 82% of cases, the symptoms had lasted for less than a week before testing.

### Diagnosis of viral upper airway infection in different specimens

Using a combined classification for all tested specimens, 157 (63%) of the 250 participants tested positive for at least one virus, with 26 of them (11%) testing positive for two or three viruses simultaneously (Fig. 2). Overall, 48% of individuals were infected with SARS-CoV-2, 5% with influenza virus (A, B, or both; hereafter influenza), and 9% were coinfected with SARS-CoV-2 and influenza. RSV was not detected in any specimen. SARS-CoV-2 and influenza A were most frequently detected in throat swabs,

TABLE 1 Demographics and clinical characteristics of the participants

| Variable | Total, $n = 250$ | Negative, $n = 93$ | SARS-CoV-2, $n = 121$ | Influenza, $n = 13$ | SARS-CoV-2 + influenza, $n = 23$ | P value[a] |
|---|---|---|---|---|---|---|
| Age | 45 [33–57] | 47 [32–57] | 45 [33–58] | 52 [35–58] | 45 [38–56] | 0.94 |
| Female | 129 (52%) | 52 (56%) | 66 (55%) | 3 (23%) | 8 (35%) | 0.05 |
| Vaccinated | | | | | | |
| SARS-CoV-2 | 230 (92%) | 85 (92%) | 115 (95%) | 11 (85%) | 19 (83%) | 0.10 |
| Influenza | 106 (43%) | 45 (49%) | 47 (39%) | 5 (39%) | 9 (39%) | 0.50 |
| Symptoms (yes) | 201 (80%) | 58 (62%) | 110 (91%) | 11 (85%) | 22 (96%) | <0.001 |
| Days from onset of symptoms | | | | | | |
| 1–3 days | 100 (50%) | 26 (45%) | 55 (51%) | 5 (46%) | 14 (64%) | 0.12 |
| 4–6 days | 63 (32%) | 15 (26%) | 34 (31%) | 6 (54%) | 8 (36%) | |
| 1–2 weeks | 30 (15%) | 13 (22%) | 17 (16%) | 0 | 0 | |
| >2 weeks | 7 (4%) | 4 (7%) | 3 (3%) | 0 | 0 | |
| Smoker | | | | | | |
| Current | 29 (16%) | 7 (10%) | 17 (19%) | 2 (20%) | 3 (19%) | 0.70 |
| Former | 29 (16%) | 13 (19%) | 12 (13%) | 1 (10%) | 3 (19%) | |
| Chronic disease | 59 (32%) | 19 (27%) | 29 (32%) | 4 (40%) | 7 (44%) | 0.55 |

[a]P value using Fisher's exact test for qualitative variables and Kruskal–Wallis test for quantitative variables. Values are numbers and percentages unless otherwise stated. Interquartile ranges are presented in brackets.

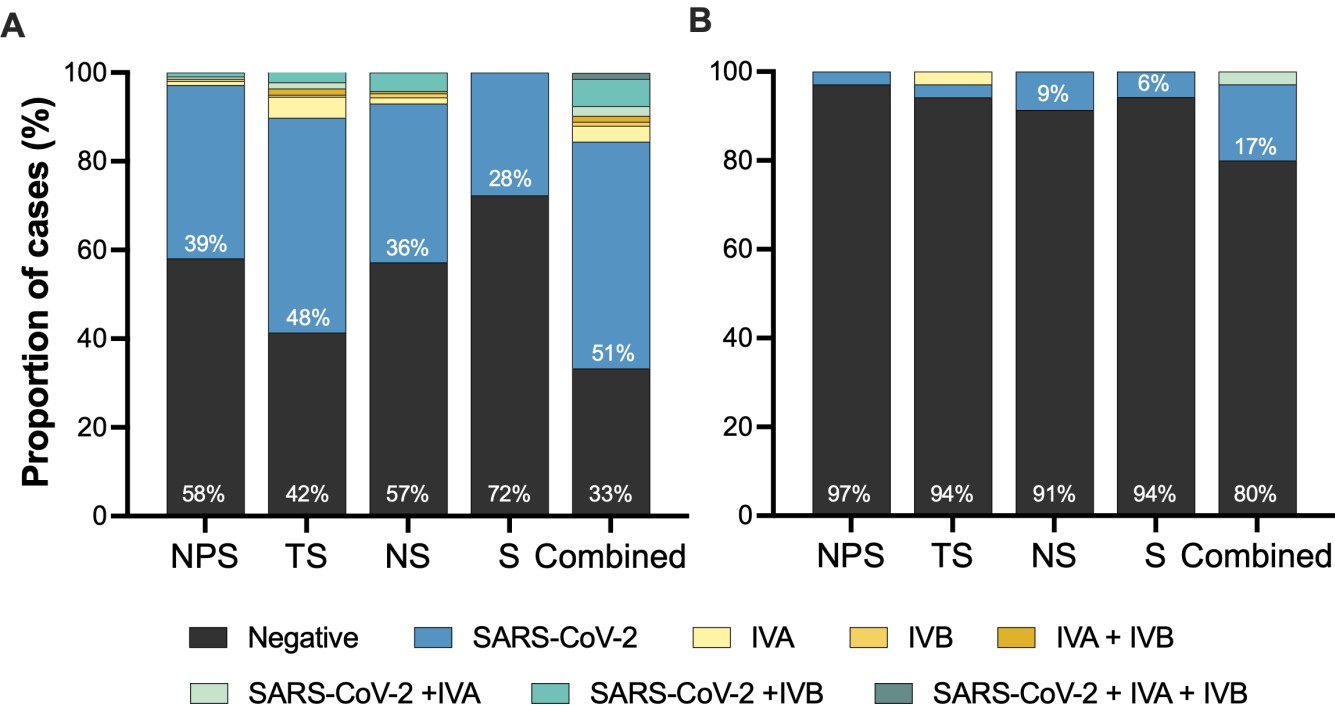

**FIG 2** Diagnosis of viral upper airway infections across specimens. Proportion of participants with a specific diagnosis in NPS, TS, NS, and saliva (S). Combined results of nasopharyngeal, throat, nasal swabs, and saliva are presented as "combined." (A) Participants requesting SARS-CoV-2 RT-PCR test for diagnosis or screening ($n = 215$) and (B) healthy individuals who were invited to participate in the study ($n = 35$). IVA, influenza virus A; IVB, influenza virus B.

whereas influenza B was most frequently detected in nasal swabs. In contrast, saliva was unsuitable for detecting either influenza A or B, although it allowed us to detect seven SARS-CoV-2 infections that were negative in other specimens. Additionally, it was not possible to process saliva samples from 16 participants due to clumping. However, the median Ct value for human RNase P in saliva (17.9, IQR: 16.6–19.3) was significantly lower than in other specimens (20.9–21.8; $P < 0.001$). Ct values for SARS-CoV-2 N1 and N2 were highly correlated in all specimens ($r_s = 0.85$–0.98; $P < 0.001$) and significantly higher (i.e., lower viral load) in saliva compared to other sites (25.1, IQR: 22.8–27.8; $P < 0.001$). Moreover, significantly lower SARS-CoV-2 N2 Ct values (i.e., higher viral load) were observed in nasopharyngeal swabs (15.3, IQR: 12.3–22.9) compared to the throat (19.2, IQR: 14.2–27.7; $P = 0.008$) and nasal swabs (19.0, IQR: 14.8–25.7; $P = 0.03$).

There were no significant differences in sensitivity for SARS-CoV-2 ($P = 0.26$) and influenza ($P = 0.62$) across different randomization sequences. However, our results showed differences in test sensitivity and NPV among sampling sites (Fig. 3). Throat swabs (79%) were significantly more sensitive than nasopharyngeal (61%), nasal swabs (62%), and saliva (43%) in detecting SARS-CoV-2 ($P < 0.001$). Likewise, the sensitivity of throat swabs to detect influenza virus (64%) was significantly higher than nasopharyngeal (17%; $P = 0.001$) but comparable to nasal swabs (42%; $P = 0.13$). Sensitivity for the detection of participants with SARS-CoV-2 and influenza was not significantly different among sampling sites ($P = 0.09$). Combining results from throat and nasal swabs increased sensitivity for SARS-CoV-2 (88%) and influenza (100%) compared to nasal swabs ($P < 0.01$). Moreover, combining results from both swabs significantly improved detection of SARS-CoV-2 and influenza (78%) compared to individual swabs (Fig. 3).

## Clinical findings and viral diagnosis

Reporting symptoms at enrollment was significantly associated with a combined positive test for any virus (OR: 6.2, 95% CI: 3.1–12.4; $P < 0.001$). Nevertheless, 62% of those who tested negative also reported symptoms. Among participants with a positive test, 86%

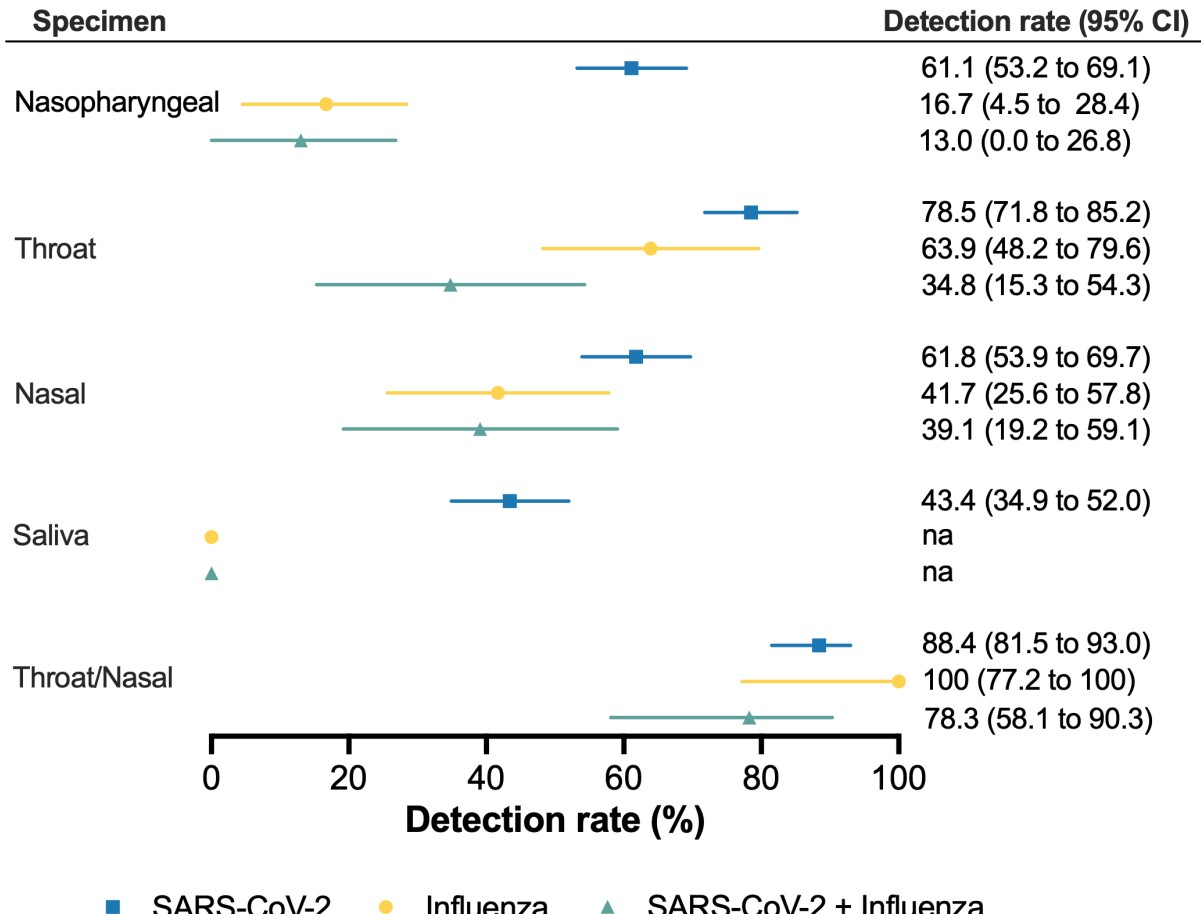

**FIG 3** Sensitivity for molecular detection of viruses in nasopharyngeal, throat and nasal swabs, and saliva specimens. Sensitivity for molecular detection of SARS-CoV-2 (n = 121), influenza (n = 13), or SARS-CoV-2 and influenza (n = 23) in nasopharyngeal, throat, nasal swabs, saliva, and a combination of throat and nasal swabs. Combined throat, nasal, nasopharyngeal, and saliva results were used as the diagnostic reference. Forest plots show sensitivity (center) and 95% confidence intervals (error bars), adjusted for randomization in specimen collection.

were tested within 6 days of symptom onset compared to 71% of negative individuals (P = 0.01). Symptoms in the upper respiratory tract, including sore throat (OR: 3.71; P < 0.001) and cough (OR: 2.20; P = 0.005), were more frequently reported at enrollment by individuals infected with SARS-CoV-2 than by those testing negative in all specimens. Hyposmia was less common but also linked to SARS-CoV-2 infections (OR: 2.50; P = 0.02). Cough was most frequent in individuals with influenza virus (OR: 4.05; P = 0.04). Systemic symptoms (headache, fever, and myalgia/arthralgia) were observed in all groups, but they were most frequent in those with SARS-CoV-2 and influenza co-infection. Most participants (86%) reported both local and systemic symptoms.

**Follow-up questionnaires**

Follow-up data were available for 189 out of the 250 participants. A total of 132 participants (70%; Table 2) reported symptoms, which lasted <2 weeks in most cases (69%). Cough was significantly less common in this group (57%) compared to those reporting a longer period of illness (80%; P = 0.01). Six out of 49 asymptomatic participants at enrollment reported symptoms after the test, but only two tested positive for SARS-CoV-2. The occurrence of incapacitating symptoms (feeling too sick to work or study) was frequent among participants with SARS-CoV-2 (66%) and SARS-CoV-2 and influenza combined (71%). Of 79 participants who reported incapacitating symptoms,

TABLE 2  Follow-up questionnaire regarding clinical characteristics of the participants

| Variable | Total, $n = 189$ | Negative, $n = 70$ | SARS-CoV-2, $n = 93$ | Influenza, $n = 10$ | SARS-CoV-2 + influenza, $n = 16$ | P value[a] |
|---|---|---|---|---|---|---|
| Symptoms before or after the test[b] | | | | | | 0.02 |
| Before and after | 109 (58%) | 31 (44%) | 62 (67%) | 6 (60%) | 10 (63%) | |
| Only before | 15 (8%) | 4 (6%) | 8 (9%) | 0 | 3 (19%) | |
| Only after | 8 (4%) | 3 (4%) | 4 (4%) | 0 | 1 (6%) | |
| Duration of symptoms[b] | | | | | | 0.49 |
| 1–3 days | 18 (14%) | 8 (21%) | 7 (10%) | 1 (17%) | 2 (14%) | |
| 4–6 days | 28 (22%) | 5 (13%) | 16 (22%) | 1 (17%) | 6 (43%) | |
| 1–2 weeks | 44 (34%) | 10 (26%) | 28 (39%) | 3 (50%) | 3 (21%) | |
| 2–4 weeks | 18 (14%) | 7 (18%) | 9 (13%) | 1 (17%) | 1 (7%) | |
| >4 weeks | 22 (17%) | 8 (21%) | 12 (17%) | 0 | 2 (14%) | |
| Requested medical attention[b] | 41 (32%) | 12 (32%) | 22 (31%) | 2 (33%) | 5 (36%) | 0.66 |
| Sickness | | | | | | 0.52 |
| Tolerable | 28 (22%) | 12 (32%) | 13 (18%) | 1 (17%) | 2 (14%) | |
| Stayed at home | 21 (16%) | 6 (16%) | 11 (15%) | 2 (33%) | 2 (14%) | |
| Incapacitating[c] | 79 (62%) | 19 (51%) | 47 (66%) | 3 (50%) | 10 (71%) | |

[a]P value using Fisher's exact test for qualitative variables and Kruskal–Wallis test for quantitative variables. Values are numbers and percentages unless otherwise stated. Interquartile ranges are presented in brackets.
[b]Among 189 who answered.
[c]Too sick to work or study, including one participant who was admitted to the hospital among the 128 participants who answered the questionnaire.

31 requested medical attention or received medical guidance over the phone. One participant with dual SARS-CoV-2 and influenza infections reported a hospital admission.

## DISCUSSION

In this study, we found that throat swabs are more sensitive than nasopharyngeal swabs for SARS-CoV-2 and influenza testing. In contrast, saliva was not suitable for detecting either influenza A or B, and it was less sensitive for the detection of SARS-CoV-2.

Our results agree with recent evidence confirming the higher sensitivity of throat swabs for SARS-CoV-2 detection by RT-PCR and rapid antigen testing at the early stages of infection (6, 10, 16, 17). SARS-CoV-2 often presented in the throat days before presenting in the nose (9, 11), where it remained up to ~10 days after a controlled inoculation (9), suggesting that the throat may play an important role during SARS-CoV-2 replication in the initial presymptomatic infection period. Although saliva has a lower sensitivity for SARS-CoV-2 infection (5, 18), probably due to the challenges with sample processing (high viscosity) or lower quality of the samples, we identified several cases with negative results in other specimens. This, together with the lower pre-analytical cost reported (5, 6), suggests that saliva specimens still can be a cost-effective alternative for SARS-CoV-2 community testing.

Centers for Disease Control and Prevention recommends a nasopharyngeal swab/aspirate as the best collection method for influenza testing (19). In contrast, we observed that throat and nasal swabs were more effective than nasopharyngeal swabs in detecting influenza. Our results are also supported by a literature review on influenza virus detection methods, which indicates that a combination of nasal and throat swabs improves the sensitivity of influenza A and B testing (20). The discrepancies in sensitivity may be due to the increased discomfort experienced by patients during nasopharyngeal collection in a community setting, which can lead to suboptimal sampling.

A strength of our study is the molecular testing conducted for four common respiratory viruses across four specimens collected from 250 participants. However, the fact that a high proportion of participants classified as negative reported symptoms suggests that the failure to detect SARS-CoV-2 or influenza in these participants likely resulted from a low viral load and shedding. Those can, in turn, depend on the current viral variant, the individual's immune status due to previous infection or vaccination,

and host-related factors such as age (21). For instance, lower SARS-CoV-2 viral load has been reported in vaccinated individuals (22, 23). Higher viral loads in SARS-CoV-2 are usually detected soon after symptom onset (24) and peak before day 5 (25). In adults with influenza, most viral shedding occurs during the first 35 daysafter illness onset (26, 27) and persists for 7 days (28), with a more variable pattern for influenza B (26). Alternatively, symptomatic participants with a negative test might have been infected with another common upper respiratory virus not tested in this study, such as rhinovirus, or their symptoms could be due to non-infectious causes such as allergies. Nonetheless, early diagnosis is essential for controlling infectious transmission within the community, and undetected cases have a potential epidemiological impact, as evidenced by recent work showing that specimens with low viral loads collected early in infection may contain infectious viruses (29).

Our study demonstrates that sensitivity to SARS-CoV-2 and influenza varies depending on the type of specimen tested. As previously reported (6), we found higher SARS-CoV-2 viral load in nasopharyngeal swabs compared to the throat swabs, but throat swabs still had a significantly higher detection. This suggests that combined swab specimens rather than a single specimen should be used to improve sensitivity, in agreement with recent studies (6, 14, 16). A SARS-CoV-2 positive specimen in 7 out of 35 healthy individuals, despite being asymptomatic before and after diagnosis, may be due to the high sensitivity of RT-PCR and could indicate the presence of viral remnants, since high Ct values were observed (31.7, IQR: 24.5–32.4). Regardless of the reason, it emphasizes the importance of combining specimens and conducting repeated tests to confirm or rule out infection, especially in individuals at higher risk of developing severe disease.

The study design allowed us to control for external confounders by randomizing the sequence of paired specimens. Collection of specimens by trained healthcare workers ensured standardized and high-quality sampling. Furthermore, the clinical information collected from participants at enrollment and through the follow-up questionnaire provided valuable information for our analysis. A limitation of this study is that sampling occurred on a single day, which reduces the likelihood of detecting infections, especially in symptomatic participants who tested negative (30). Additionally, while the healthcare workers received training, they have more experience collecting throat swabs than nasopharyngeal swabs. Future studies should consider multiple sampling days, as this would provide more informative data for guiding clinical decision-making. Furthermore, it also needs to be explored whether our findings can apply to rapid antigen testing.

In conclusion, we confirmed that throat swabs were more sensitive than nasopharyngeal swabs for molecular testing of SARS-CoV-2 and influenza. Therefore, a combined throat and nasal swab should be collected to ensure the highest test sensitivity of patients presenting with upper respiratory infection symptoms in non-hospitalized settings.

## ACKNOWLEDGMENTS

We would like to thank all the individuals who volunteered to participate in this study. We also acknowledge the healthcare workers at the Valby COVID-19 test center for their commitment in making this study possible.

This work was funded by the Novo Nordisk Foundation (grant NNF21SA0069151; T.T.). M.L.-P. was funded by Independent Research Fund Denmark (grant 0134-00123B). The funders had no role in study design, data collection and analysis, decision to publish, or preparation of the manuscript.

T.T. conceived and designed the study. T.T. was responsible for funding acquisition and project administration. T.T. and T.B. provided supervision. K.K.J., M.H.D., T.G., C.v.B., and N.K. helped plan and conduct the clinical trial. S.D.S. is responsible for data curation. H.L. and S.S.B. analyzed the samples. M.L.-P., A.K.E., H.L., and S.S.B. analyzed the data. M.L-P. drafted the manuscript. All authors edited, reviewed, and approved the final version of the manuscript.

## AUTHOR AFFILIATIONS

[1]Department of Immunology and Microbiology, Faculty of Health and Medical Sciences, University of Copenhagen, Copenhagen, Denmark

[2]Department of Clinical Medicine, University of Copenhagen, Copenhagen, Denmark

[3]Department of Infectious Diseases, Copenhagen University Hospital—Amager and Hvidovre, Hvidovre, Denmark

[4]Department of Otorhinolaryngology, Head and Neck Surgery and Audiology, Rigshospitalet—Copenhagen University Hospital, Copenhagen, Denmark

[5]Copenhagen Emergency Medical Services, University of Copenhagen, Copenhagen, Denmark

[6]National Institute of Public Health, University of Southern Denmark, Copenhagen, Denmark

[7]Centre for Diagnostics, Department of Health Technology, Technical University of Denmark (DTU), Lyngby, Denmark

[8]Department of Clinical Immunology, Rigshospitalet—Copenhagen University Hospital, Copenhagen, Denmark

[9]Copenhagen Academy for Medical Education and Simulation, Capital Region, Copenhagen, Denmark

## AUTHOR ORCIDs

Mary Lopez-Perez ⓘ http://orcid.org/0000-0002-9876-0248
Tobias Todsen ⓘ http://orcid.org/0000-0003-3267-3560

## FUNDING

| Funder | Grant(s) | Author(s) |
| --- | --- | --- |
| Novo Nordisk Fonden | NNF21SA0069151 | Tobias Todsen |

## AUTHOR CONTRIBUTIONS

Mary Lopez-Perez, Formal analysis, Visualization | Thomas Benfield, Supervision | Kathrine K. Jakobsen, Investigation | Mette Hyldig Dal, Investigation | Sabrina Dandanell Stange, Data curation | Annette Kjær Ersbøll, Formal analysis | Helene Larsen, Formal analysis, Investigation | Sanne Schou Berger, Formal analysis, Investigation | Tobias Gredal, Investigation | Christian von Buchwald, Investigation, Supervision | Nikolai Kirkby, Investigation, Supervision | Tobias Todsen, Conceptualization, Project administration, Supervision

## DATA AVAILABILITY

Data related to this study are provided in the main text. Further data supporting the findings of this study are available upon reasonable request to the corresponding author.

## ETHICS APPROVAL

The protocol was approved by the Regional Ethics Committee of the Capital Region of Denmark (protocol no. H-22022937) and the data responsible institute in the Capital Region of Denmark (protocol no. P-2022-803). The protocol was registered with the ClinicalTrials.gov database (NCT05765838). The supporting CONSORT checklist is available as supporting information (Checklist S1). Verbal and written informed consent were obtained from all participants before enrollment.

## ADDITIONAL FILES

The following material is available online.

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
