## [Reviewer comments · Microbiology Spectrum]

Microbiology Spectrum

Diagnostic performance of upper airway sampling sites for SARS-CoV-2 and Influenza testing

Mary Lopez-Perez, Thomas Benfield, Kathrine Jakobsen, Mette Dal, Sabrina Stange, Annette Ersbøll, Helene Larsen, Sanne Berger, Tobias Gredal, Christian Buchwald, Nikolai Kirkby, and Tobias Todsén

Corresponding Author(s): Tobias Todsén, Copenhagen University Hospital

Review Timeline:

Submission Date:	July 26, 2025
Editorial Decision:	August 12, 2025
Revision Received:	October 10, 2025
Accepted:	October 14, 2025

Editor: Wendy Szymczak

Reviewer(s): Disclosure of reviewer identity is with reference to reviewer comments included in decision letter(s). The following individuals involved in review of your submission have agreed to reveal their identity: Matthew Martin Hernandez (Reviewer #2)

Transaction Report:

DOI: <https://doi.org/10.1128/spectrum.02212-25>

Re: Spectrum02212-25 (Diagnostic performance of upper airway sampling sites for SARS-CoV-2 and Influenza testing: a randomised clinical trial)

Dear Dr. Tobias Todsen:

Thank you for the privilege of reviewing your work. Below you will find my comments, instructions from the Spectrum editorial office, and the reviewer comments.

Revision Guidelines

Sincerely,
Wendy Szymczak
Editor
Microbiology Spectrum

Reviewer #1 (Comments for the Author):

I am very enthusiastic about this paper and see a lot of value in it. The "major" comments need to be addressed to enhance rigor but they are easily addressable and do not change the paper. The minor comments are primarily to improve the clarity and precision of the paper to reduce potential misinterpretations and make the paper stronger.

This study reports results from a moderately sized trial whereby participants presenting for no-cost respiratory virus testing were randomized into groups with varying order of nasal, nasopharyngeal, and throat swab specimen collection by trained healthcare workers, then saliva specimen collection. Specimens collected from these participants underwent RNA extraction and RT-qPCR testing for SARS-CoV-2, Influenza and RSV. Test results were analyzed to assess how well each individual specimen type successfully detected individuals with evidence of infection in any specimen type tested.

On-the-ground experience of COVID-19 testing implementation demonstrated challenges with the feasibility of mass nasopharyngeal swab collection, and both cross-sectional and longitudinal studies of viral load and sensitivity of different specimen types have challenged whether nasopharyngeal swabs are the gold standard for detection of SARS-CoV-2, Influenza, and other upper respiratory virus infections. Since multiple common upper respiratory viral pathogens in four distinct upper respiratory anatomical sampling sites - including nasopharyngeal swabs - from each individual were evaluated, this paper presents valuable evidence to guide optimal diagnostic and screening testing strategies to reduce the spread of respiratory viruses.

Major Comments-

"We also found that participants who were vaccinated against Influenza had a significantly lower risk of developing incapacitating symptoms from the infection. Our findings, therefore, indicate that vaccination against influenza remains an important tool for reducing the risk of severe disease." And "Those vaccinated against influenza had a lower risk of developing incapacitating symptoms (odds ratio [OR] 0.5; 95% confidence interval [CI] 0.3 to 0.9; $p = 0.04$)."

The present study is not designed to assess influenza vaccine effectiveness, and underpowered for this conclusion, given only 13 participants were infected with Influenza, of whom only 10 responded to the follow up questionnaire assessing incapacitation. This analysis also did not control for confounding variables that can modulate disease severity beyond vaccination status (e.g. age, sex, medical comorbidities). Despite the p value, I suggest moderating this statement, pointing out the caveats so the readers to not over-interpret the p value or rely on it.

"Our study demonstrates the large impact on the detection rate for SARS-CoV-2 and influenza depending on the specimen collection method used."

This study did not assess different specimen collection methods; this study assessed the sensitivity of detecting individuals infected with SARS-CoV-2 and/or Influenza based on the specimen type used for testing. The Methods section describes a single collection method for each specimen type, and the authors highlight that consistency of specimen collection by healthcare workers is a strength of the study. This sentence should be revised to clarify that specimen collection method was not assessed, but the sensitivity of different upper respiratory specimen types to detect infected individuals was.

Observed clinical sensitivity of a specimen type is dependent on biological differences in how virus is shed from each anatomic sampling site over time, and the analytical sensitivity of the assay to detect virus shed from each specimen type. To support that the trends observed are driven by biological differences in viral load among specimen type and not differences in assay analytical sensitivity by specimen type, the manuscript would benefit from data demonstrating the analytical sensitivity for each specimen type used with this assay. If the analytical sensitivities for each specimen type are similar, then the results suggest biological differences in viral shedding among specimen types which would be generalizable to other assays used for testing.

Minor Comments-

"However, studies during the COVID-19 pandemic have shown that SARS-CoV-2 may be 76 detected in the oropharynx before the nasal cavity during the early stages of infection [6, 9, 10]."

The author's current reference 17 appears to support this statement as well.

"Individuals (\geq 18 years) requesting a free-of-charge SARS-CoV-2 RT-PCR test for diagnostic and screening purposes were invited to participate in the study, and not samples size was determined."

Minor grammatical errors here impact the clarity of this sentence.

"The participants were randomised in groups with different orders of nasopharyngeal, throat, and nasal swabs (Fig. 1)."

For clarity to readers, please state that the order of collection of these swab specimens was randomised, but followed by collection of saliva. (The word "collection" is missing)

"The combined results of nasopharyngeal, throat, nasal swabs and saliva tests were used as the diagnostic reference"

Could the authors state more explicitly how the participant was deemed positive or negative (or infected, not infected) based on the combination of results (e.g. negative in all specimen types classified the participant as negative, whereas a positive result in at least one specimen type classified the participant as positive)?

Sample collection

Please include the volume of viral transport media specimens were collected in, and for saliva, the target volume participants were instructed to collect. These volumes could potentially impact the relative analytical sensitivity of the assay with different specimen types.

"RT-PCR tests were considered SARS-CoV-2 positive if the cycle threshold (Ct) was 35 or lower for one or two target gene segments and below 28 for the RP gene."

Please specify, how were these Ct thresholds determined?

"The combined results of nasopharyngeal, throat, nasal swabs and saliva tests were used as the diagnostic reference to calculate sensitivity, specificity, positive predictive value (PPV), and negative predictive value (NPV) for diagnosis in each specimen. A logistic regression analysis using generalised estimating equations (GEEs) was used to compare the detection rate of SARS-CoV-2 and Influenza among specimens."

Could the authors additionally state how 95% confidence intervals are calculated on point estimates of sensitivity and "detection rate" (e.g. Figure 3)?

Could the authors please clarify how "detection rate" is calculated, and whether it is different from "sensitivity"? Please also use the terms defined consistently across the Results, Discussion, and figures (e.g. "The detection rate in all groups was improved when results from throat and nasal swabs were combined (Fig. 3).")

Also, the remainder of the manuscript only reports sensitivity and "detection rate"; specificity, PPV and NPV are not described elsewhere.

"Additionally, saliva showed more inconclusive results (12%), half of which were due to a lack of material."

Inconclusive results are mentioned here in the Results, but the criteria for classifying a sample as inconclusive is not listed in the Methods. It would seem that "inconclusive" due to a lack of material means a negative result in the human RNase P marker, but it is not clear what other criteria accounts for the other inconclusive results described here.

"3 participants were excluded due to invalid ID numbers."

It's not clear to me what this means. Does this mean the questionnaire could not be linked to test data, due to invalid ID numbers?

"Of 250 participants, 63% tested positive for at least one virus; 26 (11%) tested positive for two or three viruses (Fig. 2 and Table S1)."

Using a composite classification of all specimen types tested?

"The combined detection of SARS-CoV-2 and influenza was not significantly different among sampling sites ($p = 0.09$)."

It's not clear to me what this statement means. What is "combined detection of SARS-CoV-2 and influenza"? Is this an analysis of the 23 samples which were positive for both SARS-CoV-2 and Influenza? If so, perhaps a more clear way to describe this is "specimens from participants co-infected with both SARS-CoV-2 and Influenza".

"Our results showed significant differences in test sensitivity among sampling sites (Fig. 3 and Table S2). Throat swabs were significantly more sensitive than nasopharyngeal and nasal swabs and saliva in detecting SARS-CoV-2 (79%; $p < 0.001$). In contrast, saliva had the lowest sensitivity (43%; $p < 0.001$) compared to all specimens."

Could the authors provide for readers the sensitivities of each specimen type to detect SARS-CoV-2 here?

"The detection rate in all groups was improved when results from throat and nasal swabs were combined (Fig. 3)."

It would be valuable for the authors to provide the magnitude of improvement using combined throat and nasal sampling here, for each virus. Also, explicitly stating improvement relative to which individual specimen type, and whether that improvement

was statistically significant would be valuable.

"Reporting symptoms at enrolment was significantly associated with a positive test (OR 6.2, 95% CI 3.1 to 12.4; $p < 0.001$). Nevertheless, 62% of those who tested negative also reported symptoms."

Could the authors clarify whether this means a positive test result for any virus, in specimen type?

"Symptoms in the upper respiratory tract, including sore throat (OR 3.71; $p < 0.001$) and cough (OR 2.20; $p = 0.005$), were frequently reported at enrolment by individuals infected with SARS-CoV-2 (Fig. S3)."

Could the authors clarify that the sore throat was more frequently reported at enrolment by participants testing positive for SARS-CoV-2 in any specimen type, than individuals who did not test positive for any virus in any specimen type? This is clear from Figure S3, but without stating the reference group here in the main text, readers may interpret this to mean that sore throat was more frequently reported for SARS-CoV-2 infection relative to Influenza infection.

"In contrast, the nasal swab is performed on both sides, improving sample collection, especially in early infectious stages."

Nasopharyngeal swabs are also typically collected from both sides. (see <https://www.cdc.gov/flu-resources/media/pdfs/2024/08/flu-specimen-collection-guide.pdf>)

"A high proportion of participants (62%) with a negative test reporting symptoms suggests that these individuals may have had an undetected pathogen infection or that the testing occurred too soon to detect it."

A strength of this study design is testing for multiple common and high-consequence pathogens, in multiple sampling sites. "Undetected pathogen infection" could be understood to mean other common upper respiratory pathogens which could cause symptoms but were not tested in this study (e.g. rhinovirus, other endemic coronaviruses). I think that discussing how SARS-CoV-2 or influenza might not have been detected due to lower viral shedding or timing of testing during infection should be made more explicit, and then followed by the potential that these participants had infections with other pathogens that less commonly progress to severe disease in immunocompetent individuals, or non-infectious causes of upper respiratory symptoms (e.g. environmental irritants or allergies).

"Figure 2. Diagnosis of viral upper airway infections across specimens. Proportion of participants with a specific diagnosis in nasopharyngeal (NPS, $n = 250$), oropharyngeal (OPS, $n = 250$), nasal (NS, $n = 250$) swabs, and saliva (S, $n = 221$). All correspond to a final diagnosis regardless of the sampling site."

It's not clear to me what is meant by "All correspond to a final diagnosis regardless of the sampling site." Could the authors please clarify the definition here?

Also, the manuscript text uses the term "throat swab" to refer to collection from the posterior oropharyngeal wall and palatine tonsils, but this figure uniquely describes "oropharyngeal (OPS)" specimen type. For consistency, I encourage the authors to change the caption to make clear that OPS in the figure (and Figure S2) is referring to what is described in the main text as "throat swab".

"Four out of five (80%) individuals presented with at least one symptom, and in 82% of cases, the symptoms had lasted for less than a week before testing (Fig. S1)."

It is not clear whether the data in Figure S1 is for participants with infection, or all participants. Also, for clarity, consider revising to provide the actual n and N , or to say "four out of every five" individuals.

"Among participants with a positive test, 80% were tested within five days of symptom onset compared to 65% of negative individuals ($p = 0.04$)."

The data presented in Table 1 which demonstrates symptom onset bins of 1-3, 4-6, 7-14, >14 days does not initially suggest a difference in the distribution of test time relative to symptom onset between participants with a positive test, and those who were negative by all tests. On closer look, when infected participants are collapsed together, the distribution does appear to be more skewed towards early timing, relative to participants who did not test positive by any test. However, this is not easily apparent to readers from the data as presented in Table 1 or in Figure S1. The authors could consider adding a panel to Figure S1 that uses this cutoff of before or after five days from symptom onset to make this analysis more clear.

Figure S2

Could the authors please analyze the correlation between SARS-CoV-2 N1 and N2 Ct values, for each specimen type? The trend in Ct values for each specimen type provided in Figure S3 appears very similar between N1 and N2 targets, suggesting that the targets will correlate. Other studies have also found that these targets correlate closely. If they do correlate closely and

the same differences between specimen types are observed for both targets in this study, then separate commentary on N1 and N2 targets in the main text could be consolidated for clarity to readers. (i.e. "In contrast, the median Ct value for SARS-CoV-2 N1 gene was significantly higher (i.e., lower viral load) in saliva compared to other sites (25.1, IQR 22.8 to 27.8; $p < 0.001$; Fig. S2B). Moreover, Ct value for SARS-CoV-2 N2 was significantly lower (i.e., higher viral load) in nasopharyngeal swabs (15.3, IQR 12.3 to 22.9; Fig. S2C) compared to the throat (19.2, IQR 14.2 to 27.7; $p = 0.008$) and nasal swabs (19.0, IQR 14.8 to 25.7; $p = 0.03$).")

Reviewer #2 (Comments for the Author):

Lopez-Perez et al., report on the performance of distinct upper respiratory specimen types for diagnosing SARS-CoV-2, Influenza A/B, and RSV. Specifically, they use RT-PCR to detect viral nucleic acids in four distinct specimen types (e.g., nasopharyngeal (NPS), oropharyngeal (OPS), nasal swabs (NS), and saliva (S)) collected in six different sequential orders. While this study sheds light on the detectability of SARS-CoV-2 and Influenza nucleic acids over the course of presentation and symptom onset, the manuscript requires clarification on a number of concepts before being considered for publication:

Major Issues:

- While the authors are randomizing patients into different groups characterized by different sampling order, I caution this study is not a classical randomized clinical trial (RCT) as it's presented. Indeed, all patients undergo sampling of all four specimen types just in different sequences of sampling. One can argue that the randomization to different sequences of sampling qualifies it as an RCT; however, there is no endpoint that examines the outcomes (e.g., detection) based on the intervention (aka: the different sequence of sampling). Moreover, if the intervention was to be considered completely randomized, saliva should have been factored into the collection sequences instead of all patients providing it at the end. Rather, this is a diagnostic specimen comparison study for SARS-CoV-2/Influenza/RSV diagnostics. If the authors insist this is an RCT, they should revisit the endpoints examined in this study.
- What is the validated specimen type and method used for the public COVID-19 test center in Valby, Copenhagen, Denmark for which the center is cleared to report results? I assume this is the NPS on the RespiDetect[®] Respiratory Panel 1 RT-qPCR assay? Whatever the case, THAT should be the gold standard by which you compare everything to. This is important because the diagnostic result should be based a valid method of testing, not an alternative, non-verified/non-validated method. Indeed, you may find your alternative specimen type (e.g., OPS, NS, S) may detect virus when not detected (or lowly detected) using the conventional method. To ensure this is not a spurious, nonspecific amplification (aka: false-positive), I recommend repeating the reaction to ensure it is reproducible or testing on an orthogonal method to demonstrate consistency. Otherwise, one can argue the alternative specimen type is of inferior integrity, yielding non-specific reactions on the assay tested in the study.
- The authors state that healthy healthcare workers participated in the study (line 94-95). However, does the 253 include these healthy individuals? As it is currently written, the specific number of healthy individuals included in this study is unclear. This is important to distinguish to (1) establish a baseline of asymptomatic infection of contemporary SARS-CoV-2 and/or Influenza A/B strains and (2) to anticipate any level of non-specificity, particularly in the non-validated specimen types collected from healthy individuals. I would also create a separate panel of these healthy patients to highlight these findings better (e.g., Fig. 2A/B).
- There are more appropriate statistical tests to utilize to examine your data. For quantitative variables (e.g., age, Ct values), did you assess for normality (e.g., Shapiro-Wilk test) before knowing to use non-parametric testing (e.g., Kruskal-Wallis) instead of parametric testing (e.g., ANOVA). In addition, because you have all four specimens collected from a given sample collection event, you have the advantage of assessing for normality in Ct values and do pair-wise testing (e.g., if parametric: one-way ANOVA {plus minus} Geisser-Greenhouse correction or Hold-Sidak's multiple comparisons; if non-parametric: Friedman or Dunn's multiple comparisons).

Comment:

This paper was co-reviewed with Alex Viloría Winnett, a trainee in the lab with strong expertise in this area and the first author of many of the papers cited by the authors.

Overall Recommendation:

Modifications

(1) Are all the authors' conclusions supported by their data?

Yes if Revised

(2) Is the manuscript written in standard English and easy to comprehend?

Yes

(3) Does the study include any large datasets that need to be deposited in a public repository? If the answer is yes, please use the comment box to give us more details

No

Does the work described in this study raise any concerns about biosafety or biosecurity that should be discussed prior to publication by the ASM Responsible Publication Committee?

No

Have appropriate statistical tests been applied?

Yes

Do you want to receive recognition for this review on a Web of Science researcher profile?

No

Comments and Suggestions for the Author:

I am very enthusiastic about this paper and see a lot of value in it. The “major” comments need to be addressed to enhance rigor but they are easily addressable and do not change the paper. The minor comments are primarily to improve the clarity and precision of the paper to reduce potential misinterpretations and make the paper stronger.

This study reports results from a moderately sized trial whereby participants presenting for no-cost respiratory virus testing were randomized into groups with varying order of nasal, nasopharyngeal, and throat swab specimen collection by trained healthcare workers, then saliva specimen collection. Specimens collected from these participants underwent RNA extraction and RT-qPCR testing for SARS-CoV-2, Influenza and RSV. Test results were analyzed to assess how well each individual specimen type successfully detected individuals with evidence of infection in any specimen type tested.

On-the-ground experience of COVID-19 testing implementation demonstrated challenges with the feasibility of mass nasopharyngeal swab collection, and both cross-sectional and longitudinal studies of viral load and sensitivity of different specimen types have challenged whether nasopharyngeal swabs are the gold standard for detection of SARS-CoV-2, Influenza, and other upper respiratory virus infections. Since multiple common upper respiratory viral pathogens in four distinct upper respiratory anatomical sampling sites – including nasopharyngeal swabs – from each individual were evaluated, this paper presents valuable evidence to guide optimal diagnostic and screening testing strategies to reduce the spread of respiratory viruses.

Major Comments-

“We also found that participants who were vaccinated against Influenza had a significantly lower risk of developing incapacitating symptoms from the infection. Our findings, therefore, indicate that vaccination against influenza remains an important tool for reducing the risk of severe disease.” And “Those vaccinated against influenza had a lower risk of developing incapacitating symptoms (odds ratio [OR] 0.5; 95% confidence interval [CI] 0.3 to 0.9; $p = 0.04$).”

The present study is not designed to assess influenza vaccine effectiveness, and underpowered for this conclusion, given only 13 participants were infected with Influenza, of whom only 10 responded to the follow up questionnaire assessing incapacitation. This analysis also did not control for confounding variables that can modulate disease severity beyond vaccination status (e.g. age, sex, medical comorbidities). Despite the p value, I suggest moderating this statement, pointing out the caveats so the readers to not over-interpret the p value or rely on it.

“Our study demonstrates the large impact on the detection rate for SARS-CoV-2 and influenza depending on the specimen collection method used.”

This study did not assess different specimen collection methods; this study assessed the sensitivity of detecting individuals infected with SARS-CoV-2 and/or Influenza based on the specimen type used for testing. The Methods section describes a single collection method for each specimen type, and the authors highlight that consistency of specimen collection by healthcare workers is a strength of the study. This sentence should be revised to clarify that specimen collection method was not assessed, but the sensitivity of different upper respiratory specimen types to detect infected individuals was.

Observed clinical sensitivity of a specimen type is dependent on biological differences in how virus is shed from each anatomic sampling site over time, and the analytical sensitivity of the assay to detect virus shed from each specimen type. To support that the trends observed are driven by biological differences in viral load among specimen type and not differences in assay analytical sensitivity by specimen type, the manuscript would benefit from data demonstrating the analytical sensitivity for each specimen type used with this assay. If the analytical sensitivities for each specimen type are similar, then the results suggest biological differences in viral shedding among specimen types which would be generalizable to other assays used for testing.

Minor Comments-

“However, studies during the COVID-19 pandemic have shown that SARS-CoV-2 may be 76 detected in the oropharynx before the nasal cavity during the early stages of infection [6, 9, 10].”

The author’s current reference 17 appears to support this statement as well.

“Individuals (≥ 18 years) requesting a free-of-charge SARS-CoV-2 RT-PCR test for diagnostic and screening purposes were invited to participate in the study, and not samples size was determined.”

Minor grammatical errors here impact the clarity of this sentence.

“The participants were randomised in groups with different orders of nasopharyngeal, throat, and nasal swabs (Fig. 1).”

For clarity to readers, please state that the order of collection of these swab specimens was randomised, but followed by collection of saliva. (The word “collection” is missing)

“The combined results of nasopharyngeal, throat, nasal swabs and saliva tests were used as the diagnostic reference”

Could the authors state more explicitly how the participant was deemed positive or negative (or infected, not infected) based on the combination of results (e.g. negative in all specimen types classified the participant as negative, whereas a positive result in at least one specimen type classified the participant as positive)?

Sample collection

Please include the volume of viral transport media specimens were collected in, and for saliva, the target volume participants were instructed to collect. These volumes could potentially impact the relative analytical sensitivity of the assay with different specimen types.

“RT-PCR tests were considered SARS-CoV-2 positive if the cycle threshold (Ct) was 35 or lower for one or two target gene segments and below 28 for the RP gene.”

Please specify, how were these Ct thresholds determined?

“The combined results of nasopharyngeal, throat, nasal swabs and saliva tests were used as the diagnostic reference to calculate sensitivity, specificity, positive predictive value (PPV), and negative predictive value (NPV) for diagnosis in each specimen. A logistic regression analysis using generalised estimating equations (GEEs) was used to compare the detection rate of SARS-CoV-2 and Influenza among specimens.”

Could the authors additionally state how 95% confidence intervals are calculated on point estimates of sensitivity and “detection rate” (e.g. Figure 3)?

Could the authors please clarify how “detection rate” is calculated, and whether it is different from “sensitivity”? Please also use the terms defined consistently across the Results, Discussion, and figures (e.g. “The detection rate in all groups was improved when results from throat and nasal swabs were combined (Fig. 3).”)

Also, the remainder of the manuscript only reports sensitivity and “detection rate”; specificity, PPV and NPV are not described elsewhere.

“Additionally, saliva showed more inconclusive results (12%), half of which were due to a lack of material.”

Inconclusive results are mentioned here in the Results, but the criteria for classifying a sample as inconclusive is not listed in the Methods. It would seem that “inconclusive” due to a lack of material means a negative result in the human RNase P marker, but it is not clear what other criteria accounts for the other inconclusive results described here.

“3 participants were excluded due to invalid ID numbers.”

It’s not clear to me what this means. Does this mean the questionnaire could not be linked to test data, due to invalid ID numbers?

“Of 250 participants, 63% tested positive for at least one virus; 26 (11%) tested positive for two or three viruses (Fig. 2 and Table S1).”

Using a composite classification of all specimen types tested?

“The combined detection of SARS-CoV-2 and influenza was not significantly different among sampling sites ($p = 0.09$).”

It's not clear to me what this statement means. What is “combined detection of SARS-CoV-2 and influenza”? Is this an analysis of the 23 samples which were positive for both SARS-CoV-2 and Influenza? If so, perhaps a more clear way to describe this is “specimens from participants co-infected with both SARS-CoV-2 and Influenza”.

“Our results showed significant differences in test sensitivity among sampling sites (Fig. 3 and Table S2). Throat swabs were significantly more sensitive than nasopharyngeal and nasal swabs and saliva in detecting SARS-CoV-2 (79%; $p < 0.001$). In contrast, saliva had the lowest sensitivity (43%; $p < 0.001$) compared to all specimens.”

Could the authors provide for readers the sensitivities of each specimen type to detect SARS-CoV-2 here?

“The detection rate in all groups was improved when results from throat and nasal swabs were combined (Fig. 3).”

It would be valuable for the authors to provide the magnitude of improvement using combined throat and nasal sampling here, for each virus. Also, explicitly stating improvement relative to which individual specimen type, and whether that improvement was statistically significant would be valuable.

“Reporting symptoms at enrolment was significantly associated with a positive test (OR 6.2, 95% CI 3.1 to 12.4; $p < 0.001$). Nevertheless, 62% of those who tested negative also reported symptoms.”

Could the authors clarify whether this means a positive test result for any virus, in specimen type?

“Symptoms in the upper respiratory tract, including sore throat (OR 3.71; $p < 0.001$) and cough (OR 2.20; $p = 0.005$), were frequently reported at enrolment by individuals infected with SARS-CoV-2 (Fig. S3).”

Could the authors clarify that the sore throat was more frequently reported at enrolment by participants testing positive for SARS-CoV-2 in any specimen type, than individuals who did not test positive for any virus in any specimen type? This is clear from Figure S3, but without stating the reference group here in the main text,

readers may interpret this to mean that sore throat was more frequently reported for SARS-CoV-2 infection relative to Influenza infection.

“In contrast, the nasal swab is performed on both sides, improving sample collection, especially in early infectious stages.”

Nasopharyngeal swabs are also typically collected from both sides. (see <https://www.cdc.gov/flu-resources/media/pdfs/2024/08/flu-specimen-collection-guide.pdf>)

“A high proportion of participants (62%) with a negative test reporting symptoms suggests that these individuals may have had an undetected pathogen infection or that the testing occurred too soon to detect it.”

A strength of this study design is testing for multiple common and high-consequence pathogens, in multiple sampling sites. “Undetected pathogen infection” could be understood to mean other common upper respiratory pathogens which could cause symptoms but were not tested in this study (e.g. rhinovirus, other endemic coronaviruses). I think that discussing how SARS-CoV-2 or influenza might not have been detected due to lower viral shedding or timing of testing during infection should be made more explicit, and then followed by the potential that these participants had infections with other pathogens that less commonly progress to severe disease in immunocompetent individuals, or non-infectious causes of upper respiratory symptoms (e.g. environmental irritants or allergies).

“Figure 2. Diagnosis of viral upper airway infections across specimens. Proportion of participants with a specific diagnosis in nasopharyngeal (NPS, n = 250), oropharyngeal (OPS, n = 250), nasal (NS, n = 250) swabs, and saliva (S, n = 221). All correspond to a final diagnosis regardless of the sampling site.”

It's not clear to me what is meant by "All correspond to a final diagnosis regardless of the sampling site." Could the authors please clarify the definition here?

Also, the manuscript text uses the term “throat swab” to refer to collection from the posterior oropharyngeal wall and palatine tonsils, but this figure uniquely describes “oropharyngeal (OPS)” specimen type. For consistency, I encourage the authors to change the caption to make clear that OPS in the figure (and Figure S2) is referring to what is described in the main text as “throat swab”.

“Four out of five (80%) individuals presented with at least one symptom, and in 82% of cases, the symptoms had lasted for less than a week before testing (Fig. S1).”

It is not clear whether the data in Figure S1 is for participants with infection, or all participants. Also, for clarity, consider revising to provide the actual n and N, or to say “four out of every five” individuals.

“Among participants with a positive test, 80% were tested within five days of symptom onset compared to 65% of negative individuals ($p = 0.04$).”

The data presented in Table 1 which demonstrates symptom onset bins of 1-3, 4-6, 7-14, >14 days does not initially suggest a difference in the distribution of test time relative to symptom onset between participants with a positive test, and those who were negative by all tests. On closer look, when infected participants are collapsed together, the distribution does appear to be more skewed towards early timing, relative to participants who did not test positive by any test. However, this is not easily apparent to readers from the data as presented in Table 1 or in Figure S1. The authors could consider adding a panel to Figure S1 that uses this cutoff of before or after five days from symptom onset to make this analysis more clear.

Figure S2

Could the authors please analyze the correlation between SARS-CoV-2 N1 and N2 Ct values, for each specimen type? The trend in Ct values for each specimen type provided in Figure S3 appears very similar between N1 and N2 targets, suggesting that the targets will correlate. Other studies have also found that these targets correlate closely. If they do correlate closely and the same differences between specimen types are observed for both targets in this study, then separate commentary on N1 and N2 targets in the main text could be consolidated for clarity to readers. (i.e. “In contrast, the median Ct value for SARS-CoV-2 N1 gene was significantly higher (i.e., lower viral load) in saliva compared to other sites (25.1, IQR 22.8 to 27.8; $p < 0.001$; Fig. S2B). Moreover, Ct value for SARS-CoV-2 N2 was significantly lower (i.e., higher viral load) in nasopharyngeal swabs (15.3, IQR 12.3 to 22.9; Fig. S2C) compared to the throat (19.2, IQR 14.2 to 27.7; $p = 0.008$) and nasal swabs (19.0, IQR 14.8 to 25.7; $p = 0.03$).”)

Confidential Comments to the Editor:

None

Point-by-point response to reviewers' comments

Ref. Spectrum02212-25

Diagnostic performance of upper airway sampling sites for SARS-CoV-2 and Influenza testing: a randomised clinical trial

Mary Lopez-Perez, Thomas Benfield, Kathrine K. Jakobsen, Mette Hyldig Dal, Sabrina Dandanell Stange, Annette Kjær Ersbøll, Helene Larsen, Sanne Schou Berger, Tobias Gredal, Christian von Buchwald, Nikolai Kirkby, Tobias Todsén

Thank you for the privilege of reviewing your work. Below you will find my comments, instructions from the Spectrum editorial office, and the reviewer comments.

R/We thank the reviewers and the editor for their comments on the manuscript. Points raised by the reviewers have been addressed.

Reviewers' Comments:

Reviewer #1:

I am very enthusiastic about this paper and see a lot of value in it. The "major" comments need to be addressed to enhance rigor but they are easily addressable and do not change the paper. The minor comments are primarily to improve the clarity and precision of the paper to reduce potential misinterpretations and make the paper stronger.

This study reports results from a moderately sized trial whereby participants presenting for no-cost respiratory virus testing were randomized into groups with varying order of nasal, nasopharyngeal, and throat swab specimen collection by trained healthcare workers, then saliva specimen collection. Specimens collected from these participants underwent RNA extraction and RT-qPCR testing for SARS-CoV-2, Influenza and RSV. Test results were analyzed to assess how well each individual specimen type successfully detected individuals with evidence of infection in any specimen type tested.

On-the-ground experience of COVID-19 testing implementation demonstrated challenges with the feasibility of mass nasopharyngeal swab collection, and both cross-sectional and longitudinal studies of viral load and sensitivity of different specimen types have challenged whether nasopharyngeal swabs are the gold standard for detection of SARS-CoV-2, Influenza, and other upper respiratory virus infections. Since multiple common upper respiratory viral pathogens in four distinct upper respiratory anatomical sampling sites - including nasopharyngeal swabs - from each individual were evaluated, this paper presents valuable evidence to guide optimal diagnostic and screening testing strategies to reduce the spread of respiratory viruses.

Major Comments-

"We also found that participants who were vaccinated against Influenza had a significantly lower risk of developing incapacitating symptoms from the infection. Our findings, therefore, indicate that vaccination against influenza remains an important tool for reducing the risk of severe disease." And "Those vaccinated against influenza had a lower risk of developing incapacitating symptoms (odds ratio [OR] 0.5; 95% confidence interval [CI] 0.3 to 0.9; $p = 0.04$)."

The present study is not designed to assess influenza vaccine effectiveness, and underpowered for this conclusion, given only 13 participants were infected with Influenza, of whom only 10 responded to the follow up questionnaire assessing incapacitation. This analysis also did not control for confounding variables that can modulate disease severity beyond vaccination status (e.g. age, sex,

medical comorbidities). Despite the *p* value, I suggest moderating this statement, pointing out the caveats so the readers to not over-interpret the *p* value or rely on it.

R/We agree with the reviewer regarding the insufficient evidence for our conclusion; therefore, we decided to remove that statement to prevent misleading the reader. This is an aspect to consider when designing future studies.

"Our study demonstrates the large impact on the detection rate for SARS-CoV-2 and influenza depending on the specimen collection method used."

This study did not assess different specimen collection methods; this study assessed the sensitivity of detecting individuals infected with SARS-CoV-2 and/or Influenza based on the specimen type used for testing. The Methods section describes a single collection method for each specimen type, and the authors highlight that consistency of specimen collection by healthcare workers is a strength of the study. This sentence should be revised to clarify that specimen collection method was not assessed, but the sensitivity of different upper respiratory specimen types to detect infected individuals was.

R/ the sentence has been edited accordingly (Line 317).

Observed clinical sensitivity of a specimen type is dependent on biological differences in how virus is shed from each anatomic sampling site over time, and the analytical sensitivity of the assay to detect virus shed from each specimen type. To support that the trends observed are driven by biological differences in viral load among specimen type and not differences in assay analytical sensitivity by specimen type, the manuscript would benefit from data demonstrating the analytical sensitivity for each specimen type used with this assay. If the analytical sensitivities for each specimen type are similar, then the results suggest biological differences in viral shedding among specimen types which would be generalizable to other assays used for testing.

R/ During assay development, we determined the lowest concentration of viral RNA that can be reliably detected using serial dilutions of known quantities of viral RNA in water. However, we do not have results for each specimen type used in this study.

Minor Comments-

"However, studies during the COVID-19 pandemic have shown that SARS-CoV-2 may be 76 detected in the oropharynx before the nasal cavity during the early stages of infection [6, 9, 10]."

The author's current reference 17 appears to support this statement as well.

R/ thank you for noticing this, the reference is now incorporated in the sentence.

"Individuals (greater than or equal to 18 years) requesting a free-of-charge SARS-CoV-2 RT-PCR test for diagnostic and screening purposes were invited to participate in the study, and not samples size was determined."

Minor grammatical errors here impact the clarity of this sentence.

R/ the sentence has been edited accordingly (Line 95).

"The participants were randomised in groups with different orders of nasopharyngeal, throat, and nasal swabs (Fig. 1)."

For clarity to readers, please state that the order of collection of these swab specimens was randomised, but followed by collection of saliva. (The word "collection" is missing)

R/ the sentence was edited as suggested (Line 111).

"The combined results of nasopharyngeal, throat, nasal swabs and saliva tests were used as the diagnostic reference"

Could the authors state more explicitly how the participant was deemed positive or negative (or infected, not infected) based on the combination of results (e.g. negative in all specimen types

classified the participant as negative, whereas a positive result in at least one specimen type classified the participant as positive)?

R/Thank you for noticing this. We have now included a more detailed description of participant classification (Line 162).

Sample collection

Please include the volume of viral transport media specimens were collected in, and for saliva, the target volume participants were instructed to collect. These volumes could potentially impact the relative analytical sensitivity of the assay with different specimen types.

R/Details on sample collection were included (line 123).

"RT-PCR tests were considered SARS-CoV-2 positive if the cycle threshold (Ct) was 35 or lower for one or two target gene segments and below 28 for the RP gene."

Please specify, how were these Ct thresholds determined?

R/During assay development, we determined the lowest concentration of viral RNA that can be reliably detected. This was achieved through serial dilutions of known quantities of viral RNA in water. The Ct value corresponding to the LOD was used as the cutoff for positivity. Since the assay consistently detected viral RNA at $Ct \leq 35$, this threshold was adopted for defining a sample as positive. This cut-off is also in agreement with the recommendation provided by the supplier of the CoviDetect - COVID-19 multiplex RT-qPCR assay (PentaBase, Odense, Denmark).

Details were included in the Methods section (Line 148)

"The combined results of nasopharyngeal, throat, nasal swabs and saliva tests were used as the diagnostic reference to calculate sensitivity, specificity, positive predictive value (PPV), and negative predictive value (NPV) for diagnosis in each specimen. A logistic regression analysis using generalised estimating equations (GEEs) was used to compare the detection rate of SARS-CoV-2 and Influenza among specimens."

Could the authors additionally state how 95% confidence intervals are calculated on point estimates of sensitivity and "detection rate" (e.g. Figure 3)?

R/ Information was included in the methods section (line 173) and the Figure legend (Fig. 3).

Could the authors please clarify how "detection rate" is calculated, and whether it is different from "sensitivity"? Please also use the terms defined consistently across the Results, Discussion, and figures (e.g. "The detection rate in all groups was improved when results from throat and nasal swabs were combined (Fig. 3)").

Also, the remainder of the manuscript only reports sensitivity and "detection rate"; specificity, PPV and NPV are not described elsewhere.

R/Both terms were used as synonyms. The sensitivity term was chosen, and the document edited accordingly. Data on PPV and NPV are presented in Table S2.

"Additionally, saliva showed more inconclusive results (12%), half of which were due to a lack of material."

Inconclusive results are mentioned here in the Results, but the criteria for classifying a sample as inconclusive is not listed in the Methods. It would seem that "inconclusive" due to a lack of material means a negative result in the human RNase P marker, but it is not clear what other criteria accounts for the other inconclusive results described here.

R/The sentence was edited to reflect that 16/250 saliva samples were not tested due to clumping (Line 200).

"3 participants were excluded due to invalid ID numbers." It's not clear to me what this means. Does this mean the questionnaire could not be linked to test data, due to invalid ID numbers?

R/ the sentence has been edited accordingly (line 183).

"Of 250 participants, 63% tested positive for at least one virus; 26 (11%) tested positive for two or three viruses (Fig. 2 and Table S1)."

Using a composite classification of all specimen types tested?

R/ Yes, the sentence was edited to reflect this (line 192).

"The combined detection of SARS-CoV-2 and influenza was not significantly different among sampling sites ($p = 0.09$)."

It's not clear to me what this statement means. What is "combined detection of SARS-CoV-2 and influenza"? Is this an analysis of the 23 samples which were positive for both SARS-CoV-2 and Influenza? If so, perhaps a more clear way to describe this is "specimens from participants co-infected with both SARS-CoV-2 and Influenza".

R/ the sentence has been edited as suggested (line 225).

"Our results showed significant differences in test sensitivity among sampling sites (Fig. 3 and Table S2). Throat swabs were significantly more sensitive than nasopharyngeal and nasal swabs and saliva in detecting SARS-CoV-2 (79%; $p < 0.001$). In contrast, saliva had the lowest sensitivity (43%; $p < 0.001$) compared to all specimens."

Could the authors provide for readers the sensitivities of each specimen type to detect SARS-CoV-2 here?

R/Details about sensitivity were included (line 221).

"The detection rate in all groups was improved when results from throat and nasal swabs were combined (Fig. 3)."

It would be valuable for the authors to provide the magnitude of improvement using combined throat and nasal sampling here, for each virus. Also, explicitly stating improvement relative to which individual specimen type, and whether that improvement was statistically significant would be valuable.

R/ Information was included (line 227).

"Reporting symptoms at enrolment was significantly associated with a positive test (OR 6.2, 95% CI 3.1 to 12.4; $p < 0.001$). Nevertheless, 62% of those who tested negative also reported symptoms."

Could the authors clarify whether this means a positive test result for any virus, in specimen type?

R/ the sentence has been edited as suggested (line 232).

"Symptoms in the upper respiratory tract, including sore throat (OR 3.71; $p < 0.001$) and cough (OR 2.20; $p = 0.005$), were frequently reported at enrolment by individuals infected with SARS-CoV-2 (Fig. S3)."

Could the authors clarify that the sore throat was more frequently reported at enrolment by participants testing positive for SARS-CoV-2 in any specimen type, than individuals who did not test positive for any virus in any specimen type? This is clear from Figure S3, but without stating the reference group here in the main text, readers may interpret this to mean that sore throat was more frequently reported for SARS-CoV-2 infection relative to Influenza infection.

R/Sentence was revised for clarity (line 236).

"In contrast, the nasal swab is performed on both sides, improving sample collection, especially in early infectious stages."

Nasopharyngeal swabs are also typically collected from both sides. (see <https://www.cdc.gov/flu-resources/media/pdfs/2024/08/flu-specimen-collection-guide.pdf>)

R/ thank you for noticing this. We have now deleted the sentence.

"A high proportion of participants (62%) with a negative test reporting symptoms suggests that these individuals may have had an undetected pathogen infection or that the testing occurred too soon to detect it."

A strength of this study design is testing for multiple common and high-consequence pathogens, in multiple sampling sites. "Undetected pathogen infection" could be understood to mean other common upper respiratory pathogens which could cause symptoms but were not tested in this study (e.g. rhinovirus, other endemic coronaviruses). I think that discussing how SARS-CoV-2 or influenza might not have been detected due to lower viral shedding or timing of testing during infection should be made more explicit, and then followed by the potential that these participants had infections with other pathogens that less commonly progress to severe disease in immunocompetent individuals, or non-infectious causes of upper respiratory symptoms (e.g. environmental irritants or allergies).

R/Thank you for your positive feedback. We have now edited the paragraph in the discussion to reflect this (line 295).

"Figure 2. Diagnosis of viral upper airway infections across specimens. Proportion of participants with a specific diagnosis in nasopharyngeal (NPS, n = 250), oropharyngeal (OPS, n = 250), nasal (NS, n = 250) swabs, and saliva (S, n = 221). All correspond to a final diagnosis regardless of the sampling site."

It's not clear to me what is meant by "All correspond to a final diagnosis regardless of the sampling site." Could the authors please clarify the definition here?

R/The last column corresponds to the combined results for all tested specimens. Figure and Figure legend 2 were edited accordingly.

Also, the manuscript text uses the term "throat swab" to refer to collection from the posterior oropharyngeal wall and palatine tonsils, but this figure uniquely describes "oropharyngeal (OPS)" specimen type. For consistency, I encourage the authors to change the caption to make clear that OPS in the figure (and Figure S2) is referring to what is described in the main text as "throat swab".

R/ Figure 2 and Figure S2 were edited as suggested.

"Four out of five (80%) individuals presented with at least one symptom, and in 82% of cases, the symptoms had lasted for less than a week before testing (Fig. S1)."

It is not clear whether the data in Figure S1 is for participants with infection, or all participants. Also, for clarity, consider revising to provide the actual n and N, or to say "four out of every five" individuals.

R/ Data corresponds to all participants. The sentence (line 188) and figure legend S1 were edited accordingly.

"Among participants with a positive test, 80% were tested within five days of symptom onset compared to 65% of negative individuals (p = 0.04)."

The data presented in Table 1 which demonstrates symptom onset bins of 1-3, 4-6, 7-14, >14 days does not initially suggest a difference in the distribution of test time relative to symptom onset between participants with a positive test, and those who were negative by all tests. On closer look, when infected participants are collapsed together, the distribution does appear to be more skewed towards early timing, relative to participants who did not test positive by any test. However, this is not easily apparent to readers from the data as presented in Table 1 or in Figure S1. The authors

could consider adding a panel to Figure S1 that uses this cutoff of before or after five days from symptom onset to make this analysis more clear.

R/The sentence (Line 234) and Figure S1 were modified to better reflect the data.

Figure S2

Could the authors please analyze the correlation between SARS-CoV-2 N1 and N2 Ct values, for each specimen type? The trend in Ct values for each specimen type provided in Figure S3 appears very similar between N1 and N2 targets, suggesting that the targets will correlate. Other studies have also found that these targets correlate closely. If they do correlate closely and the same differences between specimen types are observed for both targets in this study, then separate commentary on N1 and N2 targets in the main text could be consolidated for clarity to readers. (i.e. "In contrast, the median Ct value for SARS-CoV-2 N1 gene was significantly higher (i.e., lower viral load) in saliva compared to other sites (25.1, IQR 22.8 to 27.8; $p < 0.001$; Fig. S2B). Moreover, Ct value for SARS-CoV-2 N2 was significantly lower (i.e., higher viral load) in nasopharyngeal swabs (15.3, IQR 12.3 to 22.9; Fig. S2C) compared to the throat (19.2, IQR 14.2 to 27.7; $p = 0.008$) and nasal swabs (19.0, IQR 14.8 to 25.7; $p = 0.03$).")

R/Ct values for SARS-CoV-2 N1 and N2 were highly correlated in all specimens; however, some comparisons were only significant for N2, so the sentence was edited to reflect this (line 203).

Reviewer #2

Lopez-Perez et al., report on the performance of distinct upper respiratory specimen types for diagnosing SARS-CoV-2, Influenza A/B, and RSV. Specifically, they use RT-PCR to detect viral nucleic acids in four distinct specimen types (e.g., nasopharyngeal (NPS), oropharyngeal (OPS), nasal swabs (NS), and saliva (S)) collected in six different sequential orders. While this study sheds light on the detectability of SARS-CoV-2 and Influenza nucleic acids over the course of presentation and symptom onset, the manuscript requires clarification on a number of concepts before being considered for publication:

Major Issues:

- While the authors are randomizing patients into different groups characterized by different sampling order, I caution this study is not a classical randomized clinical trial (RCT) as it's presented. Indeed, all patients undergo sampling of all four specimen types just in different sequences of sampling. One can argue that the randomization to different sequences of sampling qualifies it as an RCT; however, there is no endpoint that examines the outcomes (e.g., detection) based on the intervention (aka: the different sequence of sampling). Moreover, if the intervention was to be considered completely randomized, saliva should have been factored into the collection sequences instead of all patients providing it at the end. Rather, this is a diagnostic specimen comparison study for SARS-CoV-2/Influenza/RSV diagnostics. If the authors insist this is an RCT, they should revisit the endpoints examined in this study.

R/ Thank you for this relevant comment. We agree that this is not a typical randomized clinical trial, and to avoid confusion, we have deleted "randomized clinical trial" from the title . We randomized the order of the swabs to minimize the potential bias caused by test discomfort during the sampling, which could affect the sensitivity of the subsequent swabs. Since saliva collection was not associated with discomfort and required extra time to do it, we excluded it from the randomization process and collected it after the three swabs. We did analyze the sensitivity across different randomization sequences, although we did not find any significant differences. Moreover, sensitivity and 95% confidence intervals were calculated with adjustments to account for the order in which specimens were collected.

- What is the validated specimen type and method used for the public COVID-19 test center in Valby, Copenhagen, Denmark for which the center is cleared to report results? I assume this is the NPS on the RespiDetect[®] Respiratory Panel 1 RT-qPCR assay? Whatever the case, THAT should be the gold standard by which you compare everything to. This is important because the diagnostic result should be based a valid method of testing, not an alternative, non-verified/non-validated method. Indeed, you may find your alternative specimen type (e.g., OPS, NS, S) may detect virus when not detected (or lowly detected) using the conventional method. To ensure this is not a spurious, nonspecific amplification (aka: false-positive), I recommend repeating the reaction to ensure it is reproducible or testing on an orthogonal method to demonstrate consistency. Otherwise, one can argue the alternative specimen type is of inferior integrity, yielding non-specific reactions on the assay tested in the study.

R/ The method used for COVID-19 diagnosis at the test center was the same as in our study (CoviDetect[™], Pentabase A/S) and has been validated with both nasopharyngeal and throat swabs. Here, we used the combined results as a reference to calculate sensitivity, in line with recent studies [ref. 6, 14, 16] and CDC guidelines [<https://shorturl.at/SRRVA>], which indicate that collecting two different specimens can improve detection sensitivity. Thus, we believe our approach is valid, and the results represent true positives.

- The authors state that healthy healthcare workers participated in the study (line 94-95). However, does the 253 include these healthy individuals? As it is currently written, the specific number of healthy individuals included in this study is unclear. This is important to distinguish to (1) establish a baseline of asymptomatic infection of contemporary SARS-CoV-2 and/or Influenza A/B strains and (2) to anticipate any level of non-specificity, particularly in the non-validated specimen types collected from healthy individuals. I would also create a separate panel of these healthy patients to highlight these findings better (e.g., Fig. 2A/B).

R/Yes, the total includes 35 healthy individuals. This sentence was modified accordingly (Line 185) and discussed (Line 322). A new Figure 2 showing those data is included.

- There are more appropriate statistical tests to utilize to examine your data. For quantitative variables (e.g., age, Ct values), did you assess for normality (e.g., Shapiro-Wilk test) before knowing to use non-parametric testing (e.g., Kruskal-Wallis) instead of parametric testing (e.g., ANOVA). In addition, because you have all four specimens collected from a given sample collection event, you have the advantage of assessing for normality in Ct values and do pair-wise testing (e.g., if parametric: one-way ANOVA {plus minus} Geisser-Greenhouse correction or Hold-Sidak's multiple comparisons; if non-parametric: Friedman or Dunn's multiple comparisons).

R/Thank you for your comment. Normality was tested before choosing the corresponding statistical test. Regarding paired testing, this is not possible due to the different proportions of positive samples for each specimen.

Re: Spectrum02212-25R1 (Diagnostic performance of upper airway sampling sites for SARS-CoV-2 and Influenza testing)

Dear Dr. Tobias Todsén:

Your manuscript has been accepted, and I am forwarding it to the ASM production staff for publication. Your paper will first be checked to make sure all elements meet the technical requirements. ASM staff will contact you if anything needs to be revised before copyediting and production can begin. Otherwise, you will be notified when your proofs are ready to be viewed.

Sincerely,
Wendy Szymczak
Editor
Microbiology Spectrum